# The Dynamic Nexus of Fossil Energy Consumption, Temperature and Carbon Emissions: Evidence from Simultaneous Equation Model

**DOI:** 10.3390/ijerph20032042

**Published:** 2023-01-22

**Authors:** Chengtao Deng, Zixin Guo, Xiaoyue Huang, Tao Shen

**Affiliations:** 1Postdoctoral Workstation of Guangxi Rural Credit Union and Southwestern University of Finance and Economics, Nanning 530000, China; 2School of Business, Zhengzhou University, Zhengzhou 450001, China; 3School of Economics and Management, Nanning Normal University, Nanning 530001, China; 4The Institute for Sustainable Development, Macau University of Science and Technology, Macao 999078, China

**Keywords:** fossil energy consumption, temperature, carbon emissions, sustainable economic growth, international competitiveness

## Abstract

With the continuous increase in global fossil energy consumption, carbon dioxide emissions and the greenhouse effect have gradually increased. This study uses a simultaneous equations model to explore the dynamic nexus of fossil energy consumption, temperature, and carbon emissions in OECD and non-OECD countries, with panel data from 2004 to 2019. The results show that the improvement of international competitiveness has reduced the frequency of extreme weather in OECD and non-OECD countries, significantly reducing fossil energy consumption in non-OECD countries and carbon emissions in OECD countries. Sustainable economic growth has significantly reduced fossil energy consumption in OECD countries but increased carbon emissions, especially in non-OECD countries. In addition, in the short term, the improvement of international competitiveness has significantly reduced fossil energy consumption and carbon emissions in OECD and non-OECD countries. In the long term, the improvement of international competitiveness has a greater impact on reducing fossil energy consumption and carbon emissions in non-OECD countries and has a significant impact on reducing the frequency of extreme weather in OECD countries. Moreover, the long-term impacts of sustainable economic growth on fossil energy consumption and carbon emissions are more significant.

## 1. Introduction

The continuous growth of fossil energy consumption around the world not only increases carbon emissions and aggravates environmental pollution but also leads to frequent extreme weather, such as cold waves in the United States, high temperatures in Europe, and polarization of droughts and floods in China [1]. A higher level of economic development may lead to higher energy consumption, leading to climate change. The increase in urbanization and transportation also increases the level of carbon emissions [2]. Therefore, it has become a global consensus to reduce fossil energy consumption, mitigate abnormal climate change and improve the ecological environment [3]. At present, the economic growth rate and international competitiveness level of OECD countries are generally higher than that of non-OECD countries [4]. Driven by high income, the growth rate of energy consumption for temperature regulation in OECD countries may be higher than that of non-OECD countries, but the utilization rate of renewable energy and energy efficiency is relatively high, and the environmental pollution is relatively low [5]. In this context, this study examines the dynamic nexus of fossil energy consumption, temperature, and carbon emissions, which will help non-OECD countries to improve environmental quality through sustainable economic growth.

The research areas include energy consumption, climate change, and environmental pollution, which are hot issues at present. Most existing studies show that the energy consumption of countries at different stages of development to cope with climate change varies greatly [6]. OCED members are mainly developed countries, which are at a stage of high economic development, advanced technology, and high living standards. In other words, compared with non-OECD countries, OECD countries usually consume more energy to regulate temperature. Fossil energy consumption is the main source of carbon emissions, and sustainable economic growth should reduce the adverse impact of carbon emissions on the environment; otherwise, it will damage a country’s international competitiveness [7]. The proposal of policies such as carbon peak, carbon neutralization, and environmental regulation not only promote the adjustment of energy structure, improve climate change and alleviate environmental pressure but also improve international competitiveness, which helps to achieve sustainable economic growth [8].

The framework of energy–climate–environment includes climate change, energy consumption, and environmental pollution three functions [9]. The cross-validation shows that the three equations should not be studied separately, which confirms that the simultaneous equations can effectively estimate the relationship between the energy–climate–environment, and the simultaneous equations model is an effective method to explore the dynamic nexus of energy–climate–environment [10]. The simultaneous equation set model can simultaneously include the equations of climate change, energy consumption, and environmental pollution. It can not only simultaneously examine the internal feedback mechanism between climate change, energy consumption, and environmental pollution but also facilitate a more comprehensive examination of the impact of exogenous variables of the system according to the empirical analysis results. According to the existing literature, the research areas of this study include fossil energy consumption, temperature, and carbon emissions from the perspective of sustainable economic growth and international competitiveness through the simultaneous equations model.

The contribution of this study is the introduction of sustainable economic growth and international competitiveness, which have never been simultaneously included in the energy–climate–environment framework. The simultaneous equation model can effectively estimate the dynamic nexus of fossil energy consumption, temperature, and carbon emissions, which is helpful in drawing reliable empirical results. In addition, this study adopts seven dimensions and twelve indicators, respectively, and uses the principal component analysis to calculate economic growth and international competitiveness, as well as uses the average temperature of three months in winter and summer to measure the temperature change, in order to more comprehensively and systematically measure the level of the three variables. Moreover, considering the sample intervals heterogeneity, this study examines the dynamic nexus of fossil energy consumption, temperature, and carbon emissions in the short term (2015–2019) and in the long term (2004–2019). To sum up, based on panel data of thirty-five OECD and thirty-six non-OECD countries in the world from 2004 to 2019, this study estimates the response differences of fossil energy consumption to temperature under different economic growth rates and international competitiveness levels, which could assess the ability of countries to deal with abnormal climate change and carbon emission reduction. These results are of great significance for OECD and non-OECD countries to formulate energy conservation and emission reduction strategies, develop a low-carbon economy, promote sustainable economic growth, and enhance international competitiveness.

## 2. Literature Review

### 2.1. The Energy–Climate–Environment Nexus

Energy–Climate–Environment nexus studies the causality among energy consumption, climate change, and carbon emissions. Energy is a basic input to carry out economic activities [11] and is described as a major source of greenhouse gas emissions. Owd [12] proved that 73.2% of global greenhouse gas emissions come from energy consumption, and most studies have shown that greenhouse gas emissions and temperature changes have a strong correlation [13]. The study of the energy–climate–environment nexus is of great significance to the mitigation of global warming and abnormal climate change, low carbon economy, and the establishment of a sustainable energy system [10]. According to the similarity of research objects in previous literature, there is a degree of correlation between energy, climate, and the environment.

In recent decades, the energy–climate–environment nexus has been a subject of academic research. There are three different research branches in the literature to explore the relationship between target variables. The first branch focuses on the relationship between energy consumption and climate change. Extreme temperature changes could significantly increase energy consumption [14]. For example, extreme temperatures in winter and summer will increase the demand for electricity, thereby increasing energy consumption [15]. Energy consumption has an adverse impact on the environment in most countries, leading to more extreme climates [16]. Some scholars also say that if the proportion of renewable energy is increased, the negative impact of energy consumption on climate change will be reduced [17]. On the other hand, the formulation and implementation of climate policies would also affect the energy demand of various industries, especially energy-intensive industries [18]. The second branch investigates the relationship between climate change and carbon emissions. There are synergies and co-benefits between climate and air quality [19]. Huge amounts of greenhouse gases might result in serious and irreversible outcomes for the whole climate system [20]. The third branch is related to energy consumption and carbon emissions. There is a consensus that energy consumption is a main source of carbon emissions.

### 2.2. Sustainable Economic Growth and the Energy–Climate–Environment

In terms of the relationship between economic growth and energy, since the pathbreaking study of Kraft [21], the causality between energy consumption and economic growth has been the focus of discussion among studies [22]. Energy is one of the important input factors of economic growth, and the relationship between them directly affects the economic policy and energy consumption policy of a region. However, in existing studies, scholars have different views on the relationship between these two variables, which could be mainly divided into four hypotheses [23]. Most scholars believe that energy consumption will significantly stimulate a country’s economic growth, supporting the growth hypothesis [24]; Some scholars believe that energy consumption and the economy interact with each other and have the same trend of change, supporting the feedback hypothesis [25,26]; Nasreen et al. [27] proposed that energy conservation policies aimed at reducing energy consumption may not adversely affect economic growth, supporting the conservation hypothesis; Sunde [28] believe that there is no significant causal relationship between energy consumption and economic growth, supporting the neutrality hypothesis.

In terms of the relationship between economic growth and climate, extreme climate change has been widely proven to harm economic growth in many ways [29]. Extreme climate change, such as natural disasters such as typhoons and floods, would cause direct economic losses in the affected areas, which is not conducive to economic growth [30]. Moreover, extreme temperatures and droughts would be detrimental to agricultural production, thus affecting economic growth [31]. Some scholars also found that extreme climate would reduce the GDP growth rate by reducing the productivity of labor and capital factors [32].

For studies discussing the relationship between the environment and the economy, most of the existing literature relies on the environmental Kuznets curve (EKC) [26]. Initially, Grossman and Krueger [33] confirmed the inverted U-shaped relationship of the EKC curve; that is, environmental pollution increases with the increase in per capita GDP at the beginning, and environmental pollution declines after per capita GDP reaches a certain threshold. Some studies affirm the EKC curve, such as Nasir and Ur-Rehman [34] and Saboori et al. [35] confirmed the existence of the EKC curve by examining the relationship between carbon emissions and incomes in Malaysia and Pakistan, respectively. Moreover, Liu et al. [36] focus on South African countries with mineral resources as the main economic source, proving that the economic growth of South African countries will lead to an increase in regional carbon dioxide emissions.

### 2.3. International Competitiveness and the Energy–Climate–Environment

Compared with the previous literature on economic growth, it is a new perspective to introduce international competitiveness into the simultaneous equation of energy–climate–environment nexus. International competitiveness is composed of economic, health, education, infrastructure, and other indicators, reflecting a country’s ability to create added value and maintain national wealth growth. International competitiveness and economic growth promote each other [37]. Compared with economic growth, it is a new perspective to introduce international competitiveness into the simultaneous equation of the energy–climate–environment nexus. In fact, there is also a clear interaction between international competitiveness and energy consumption and carbon emissions (see Figure 1). Shuai et al. [38] suggested that the country should not rely too much on energy consumption and should not rely only on energy-intensive industries to develop the economy; otherwise, it will damage international competitiveness. Luìs and Gabriel [39] used the Kaleckian growth model to prove that increased energy efficiency indirectly leads to an increase in international competitiveness.

From the perspective of international competitiveness and climate research, Ward et al. [40] asserted that climate policies could affect the competitiveness of specific industries, economic activities, and jobs in countries, but the effects are highly variable. The international competitiveness of developed countries is likely to improve, while that of emerging countries may be seriously adversely affected. OCED members are mainly developed countries, which are at a stage of high economic development, advanced technology, and high living standards. Compared with non-OECD countries, OECD countries usually consume more energy to regulate temperature (see Figure 2). Antimiani et al. [8] considered that the EU has a unilateral climate policy to mitigate climate change by reducing carbon emissions and promoting low-carbon technologies. This policy risks creating distorting effects on a global scale, with a significant impact on international competitiveness. However, investing in decarbonization strategies for energy efficiency and renewable energy from a long-term perspective would not only protect vulnerable manufacturing sectors but also enhance the international competitiveness of technologically advanced industries.

To sum up, based on the collation and summary of the above references, international competitiveness and sustainable economic growth are of great significance to the study of the energy–climate–environment framework. Therefore, this study estimates the difference in impacts of fossil energy consumption on temperature under different economic growth rates and international competitiveness levels in OECD countries and non-OECD countries, respectively, and further evaluates the ability of countries to cope with abnormal climate change and carbon emission reduction. Finally, the research purpose of this study is to draw empirical results that would help countries formulate effective energy and environmental policies to cope with abnormal climate change, reduce carbon emissions, improve international competitiveness and achieve sustainable economic growth.

## 3. Data and Methods

Since the energy–climate–environment relationship covers the causal nexus between energy consumption, climate change, and environment pollution [41]. The estimation method should not only determine the three variables but also allow the reverse causality. Therefore, this study uses the simultaneous equation model to solve this problem.

### 3.1. Climate Change Function

Climate change has attracted more countries’ attention in recent years [42]. This study regards international competitiveness as a factor. However, the international competitiveness index is rarely included in the nexus, which reflects a country’s competitiveness level. Therefore, the climate function is as follows:(1)$$temp_{it}=\alpha_{0}+\alpha_{1}k_{it}+\alpha_{2}e_{it}+\alpha_{3}GCI_{it}+\varepsilon_{1,it}$$
where *temp* is the difference between summer temperature and winter temperature. In the southern hemisphere, the opposite is true. *k* is the capital per capita; *e* is the proportion of fossil fuel energy consumption in total energy consumption.

### 3.2. Energy Consumption Function

Based on previous studies, international competitiveness and sustainable economic growth are included in the energy consumption function as follows:(2)$$e_{it}=\beta_{0}+\beta_{1}temp_{it}+\beta_{2}ind_{it}+\beta_{3}GCI_{it}+\beta_{4}seg_{it}+\varepsilon_{2,it}$$
where *ind* is industrialization, *GCI* is the international competitiveness index and *seg* is the sustainable economic growth index, which is calculated by Principal Component Analysis (PCA) and consists of seven indexes, as shown in Table 1.

### 3.3. Environmental Pollution Function

To study the impact of international competitiveness and sustainable economic growth on carbon emissions, international competitiveness and sustainable economic growth have been included in the environmental pollution function as follows:(3)$$pol_{it}=\gamma_{0}+\gamma_{1}temp_{it}+\gamma_{2}temp_{it}^{2}+\gamma_{3}e_{it}+\gamma_{4}GCI+\gamma_{5}seg_{it}+\gamma_{7}urb_{it}+\gamma_{8}poli+\varepsilon_{3,it}$$
where *pol* denotes carbon emissions per capita; $temp^{2}$ denotes *temp* squared; *urb* stands for urbanization; *poli* stands for climate policy. Moreover, *ε* is the error term. Climate change variables are not logarithmic.

From Equations (4)–(6), a three-dimensional simultaneous equation has been developed for analyzing the energy–climate–environment nexus. In summary, the structural equation is as follows:(4)$$temp_{it}=\alpha_{0}+\alpha_{1}k_{it}+\alpha_{2}e_{it}+\alpha_{3}GCI_{it}+\varepsilon_{1,it}$$
(5)$$e_{it}=\beta_{0}+\beta_{1}temp_{it}+\beta_{2}ind_{it}+\beta_{3}GCI_{it}+\beta_{4}seg_{it}+\varepsilon_{2,it}$$
(6)$$pol_{it}=\gamma_{0}+\gamma_{1}temp_{it}+\gamma_{2}temp_{it}^{2}+\gamma_{3}e_{it}+\gamma_{4}GCI+\gamma_{5}seg_{it}+\gamma_{7}urb_{it}+\gamma_{8}poli+\varepsilon_{3,it}$$

### 3.4. Variable Selection

(1)Dependent variable

Although many countries greatly develop renewable energy, most countries in the World Bank are still dominated by fossil energy, so this study chooses fossil energy consumption as the proportion of total energy consumption to define energy consumption variables. In terms of environmental pollution, some scholars use sulfur dioxide, industrial wastewater, and so on to represent environmental pollution [43], and more scholars use carbon dioxide emissions. Since carbon dioxide is still the main emission of greenhouse gas [44], this study uses carbon emissions to represent environmental pollution, and the average temperature of three months in summer or winter is used as the climate change, which can better reflect the change of extreme temperature [45]. The proportion of fossil energy consumption and carbon emissions come from the World Bank, respectively, while climate change comes from National Centers for environmental information (NOAA).

(2)Independent variable

This study selects the Global Competitiveness Index (GCI) issued annually by the World Economic Forum to represent global competitiveness [46]. This competitiveness is continuously compared with the competitiveness development of other economies or regions through the sustainable improvement of living standards and employment and other macroeconomic indicators in order to determine the competitiveness stage of economies in the global economy and to verify the view that the larger the economic scale, the stronger the competitiveness. GCI consists of twelve competitiveness pillar projects, including economy, health, education, infrastructure, and other aspects. It provides a comprehensive picture for identifying the competitiveness of countries in the world at different stages of development. In addition, in terms of economic growth, many scholars use sustainable economic growth to represent economic growth [47,48]. Based on the experience of Khan [49], this study selects seven dimensions, including Trade, Agriculture, forestry, and fishery, Value added, Population growth, Taxes on income, Profits, and capital gains, Inflation, Consumer prices, Final consumption expenditure, Exports of goods and services. The PCA is used to reduce the dimensions of index construction so as to build indicators of sustainable economic growth. The data on the economic growth variables are from the World Bank.

(3)Control variable

This study introduces the dummy variable, namely climate policy, as the control variable. If the country has signed the Kyoto Protocol or the Paris Agreement, it will be recorded as 1, otherwise recorded as 0. In addition, it is obvious that industrialization rate, population, and urbanization rate have an impact on economic growth, energy consumption, climate change, and environmental pollution [50,51]. To improve the accuracy of model estimation, they are used as control variables.

### 3.5. The Estimation Method

As the simultaneous equations are constructed, there are two choices based on estimation, one is the difference estimation, and the other is the system estimation. Under the energy-environment-growth nexus, the simultaneous equation is much more efficient than the single equation, but care needs to be taken to specification of the simultaneous equation [52]. In order to solve the endogenous problem, it is necessary to find external tool variables. However, it is difficult to find external tool variables due to changes in units and time. The Generalized Method of Momnets (GMM) can effectively use its internal tool variables to solve the above problem. With the energy-environment-growth nexus, Saidi and Hammami [53] and Sekrafi and Sghaier [54] used differenced-GMM in their studies, and Adewuyi and Awodumi [55] used system-GMM in the interrelationship of energy-environment-growth. Therefore, this study chooses to apply the differenced-GMM and system-GMM estimation, respectively, in the energy–climate–environment nexus.
(7)$$y_{\mathrm{it}}=x_{\mathrm{it}}\beta +\varphi y_{i,t-1}+c_{i}+\varepsilon_{\mathrm{it}}$$
where *t* represents time, and *i* represents the cross-section units (countries). The error term consists of the fixed individual effect c*_i_* and idiosyncratic shock *ε_it_*. Attributes of fixed individual effects and idiosyncratic shocks are attributed to the following equation:(8)$$E\left(c_{i}\right)=E\left(\varepsilon_{it}\right)=E\left(c_{i}\varepsilon_{it}\right)=0$$

By taking the difference from Equation (8) to remove the individual effects c*_i_*, the following equation as follow:(9)$$∆y_{\mathrm{it}}=(∆x{)}_{\mathrm{it}}\beta +\varphi (∆y_{i,t-1})+∆\varepsilon_{\mathrm{it}}$$
where ∆ represents the first-order difference. The predetermined variables are endogenous in first difference, but the deeper lag of the regression variable is still an effective instrument of the GMM framework. The differenced-GMM method uses the lagged and endogenous variables as instrument variables in the first difference, while the system-GMM approach integrates the original equation into the first difference equation. In addition, the endogeneity of GMM estimation instruments and the first difference lag endogeneity of the predeterminate variable are discussed, which shows that individual effect is not related to disturbance terms. Under the condition of heteroscedasticity, the Hansen test with excessive recognition restriction was used to examine the instruments’ validity.

### 3.6. Data Source

This study uses panel data, which can better explain the heterogeneity between countries and reduce the relatively short time series of data. This study selects seventy-one countries during 2004–2019 and divides them according to whether they are OECD countries, including thirty-five OECD countries and thirty-six non-OECD countries, so as to analyze them in groups and compare the differences and links between them. In addition, Table 2 shows the descriptive statistics of the data in OECD and non-OECD countries.

## 4. Results

### 4.1. The Results of Climate Change Function

The study shows the impact of competitiveness on climate change in OECD and non-OECD countries through differenced-GMM and system-GMM, respectively. The results of the climate change function are shown in Table 3 and Table 4. For OECD countries, the coefficients of the GCI are 0.228 and 0.106, respectively. For non-OECD countries, the coefficients of the GCI are −0.0633 and −0.389, respectively. They indicate the importance of competitiveness as a positive factor in OECD countries, but the impact of competitiveness on the climate function is negative in non-OECD countries. Surprisingly, competitiveness has a great difference impact on OECD and non-OECD. Therefore, this study infers that when competitiveness improves, the impact of OECD countries on environmental nature is obviously more significant due to their large size and high productivity. However, non-OECD countries, due to their weak overall level, when their competitiveness increases, will reduce the extensive development of resources, shift to more reasonable development methods, and even begin to increase the development of tertiary industries. These could explain the difference in competitiveness on climate change between OECD and non-OECD countries. As for the Hansen test, it presents the instruments were valid at a 5% risk level. The results of the Arellano–Bond test also indicate that the estimators are consistent.

### 4.2. The Results of Energy Consumption Function

The impact of competitiveness and sustainable economic growth on energy consumption is divided into OECD countries and non-OECD countries, which are shown in Table 5 and Table 6. Based on the results of the energy consumption function provided by both OECD and non-OECD countries, the temperature difference has a negative effect on energy consumption as a whole. The greater the temperature difference, the more extreme cold or hot weather, and the more dependent on energy. The coefficients of GCI for energy consumption are −0.00898, −0.00649, −0.0100, and −0.00712 at the significant level of 5% in non-OECD countries, but the coefficients for OECD countries are not significant. It can be seen that the impact of competitiveness on energy consumption shows a significant negative effect in non-OECD countries, but it is not significant in OECD countries. Obviously, due to the poor strength base of non-OECD countries, when competitiveness changes, they are more sensitive to energy consumption, while OECD countries are less sensitive to energy consumption because of their strong national strength. In addition, the coefficients of seg for energy consumption are −0.256, −0.213, and −0.187 at the significant level of 1% in OECD countries, and the coefficients for non-OECD countries are −0.110 at the 10% significant level. It shows that in terms of sustainable economic growth, OECD countries and non-OECD countries show a significant negative correlation, indicating that when the level of sustainable economic growth increases, fossil energy consumption will be significantly reduced. There is no doubt that the reduction of fossil energy use and carbon emissions is becoming particularly important for all countries now and in the future. Therefore, the above analysis is of great significance for OECD countries and non-OECD countries to formulate policies. Finally, the Hansen test showed that it was effective at the 5% level.

### 4.3. The Results of Environmental Pollution Function

Table 7 and Table 8 show the results of climate change, sustainable economic growth, and competitiveness on environmental pollution in OECD and non-OECD countries. For OECD and non-OECD countries, the temperature difference has a negative impact on carbon emissions. However, the square of temperature difference has a positive impact on carbon dioxide. That is, if the temperature difference is within the threshold range, its increase will reduce carbon emissions. When the temperature difference exceeds the threshold, the increase in temperature difference will increase carbon emissions. The changing trend and temperature difference threshold of the two areas are similar and gradually tend to be stable and gentle. Competitiveness has a significant negative effect on energy consumption in OECD countries, while it has the opposite effect in non-OECD countries. This is because when OECD countries become more competitive, they will pay more attention to the control of carbon dioxide emissions, but for non-OECD countries, they will increase industrial production and do not have enough technology to control pollutant emissions. In both OECD and non-OECD countries, there is a significant positive correlation between economic growth and carbon dioxide emissions, which is obviously in line with common sense.

However, according to the EKC hypothesis, when the economy grows to a certain extent, it will have a negative effect on carbon dioxide emissions [51]. Finally, according to the results of Table 7 and Table 8, the number of countries was significantly greater than the number of instrumental variables, and the Hansen test showed that the instrumental variables were effective at a risk level of 5%.

### 4.4. Time Heterogeneity Analysis

In order to reflect the important impact of international competitiveness and sustainable economic growth in energy–climate–environment nexus from the perspective of time, this study divides it into two stages and presents them in Table 9 and Table 10. For OCED countries, the regression coefficients of GCI for climate change are 0.675 and 0.228 in the short and long term, respectively, which shows that the competitiveness of OECD countries has a positive impact on climate change in the short and long term. For non-OCED countries, the coefficients of seg for climate change are −0.595 and −0.0633 in the short and long term, respectively, which indicates that the competitiveness of non-OECD countries has a negative impact on climate change in the short and long term. In OECD countries, the coefficients of the GCI for energy consumption are −0.0094 and −0.0027 at the 10% significant level in the short term, but the coefficients of the seg are −0.365 and −0.187 at the 1% significant level in the long term. It shows that the impact of competitiveness on energy consumption is negative in the short term, and sustainable economic growth is negative in the long period. For non-OECD countries, the coefficients of the GCI for energy consumption are −0.0078 and −0.0064 at the 10% significant level in the short term and are −0.0100 and −0.00712 at the 5% significant level in the long term. The coefficients of the seg are −0.110 at the 10% significant level in the long term. It can be seen that the impact of competitiveness on energy consumption is negative in every period. However, sustainable economic growth is different in the long-term and short-term. The long-term sustainable economic growth has a negative correlation to energy consumption.

However, the relationship between sustainable economic growth in the short term is not significant. In the pollution function, there is a contradiction between the different GMM and the system GMM results. As the system-GMM is more optimized, the system-GMM results are mainly used in this study. For OECD countries, competitiveness has a significant negative correlation with the environmental pollution in the long and short term. Sustainable economic growth has no significant impact on environmental pollution. For non-OECD countries, both competitiveness and sustainable economic growth have a significant positive correlation with the environmental pollution in the long and short term.

### 4.5. Discussion

In sum, international competitiveness and sustainable economic growth are important variables that affect climate change, energy consumption, and environmental pollution, especially in the long term is more significant than short term, which is similar to the results of Rahman [11] and Shang et al. [14], they believe that energy is the basic input of economic activities, and economic growth would have promoted the demand for renewable energy. Whether for OECD countries or non-OECD countries, international competitiveness and sustainable economic growth are extremely important, and the impact on climate change, energy consumption, and environmental pollution has a certain time lag effect. In the short term, the improvement of international competitiveness and sustainable economic growth on climate change, the reduction of energy consumption and environmental pollution may not be obvious, but through long-term development, it will have a significant impact, which is different from the results of Destek [25] and Wang [26], they hold that the interaction between energy consumption and economic growth has the same trend of change. Therefore, this has a certain practical significance for policy formulation in OECD countries and non-OECD countries.

## 5. Conclusions and Policy Implications

The empirical results have three new findings. Firstly, energy consumption has a significant positive correlation with economic growth and is the main driving force of economic growth. Fossil energy consumption promotes carbon emissions, which is particularly prominent in non-OECD countries with fast economic growth. Secondly, the impacts of international competitiveness on temperature change are obviously different in OECD and non-OECD countries. For OECD countries, the impact of international competitiveness on temperature change is positive. Due to the improvement of international competitiveness is mainly achieved through scientific and technological progress, which is often accompanied by the increase in clean energy consumption, the improvement of energy efficiency and the reduction of carbon emissions. However, for non-OECD countries, the impact of international competitiveness on temperature change is negative because non-OECD countries mainly develop heavy industry to improve international competitiveness, which is usually accompanied by a large amount of energy consumption and carbon emissions. Thirdly, compared with OECD countries, the impact of improving international competitiveness on reducing energy consumption and carbon emissions is more significant in non-OECD countries. This is because the improvement of competitiveness could reduce the utilization rate of fossil energy in energy consumption, which is conducive to increasing the proportion of clean energy consumption.

The results of this study would help governments around the world formulate reasonable and effective energy and environmental policies to deal with extreme weather change, reduce carbon emissions and achieve sustainable economic growth. Specifically, decision-makers could use temperature methods (such as summer and winter) to measure climate change and estimate its relationship with energy consumption and carbon emissions. Then, decision-makers could establish a dynamic monitoring system of climate change, energy consumption, and carbon emissions in order to identify the changes in energy consumption and carbon emissions in hot summer and cold winter, and predict the future energy consumption and carbon emissions, so as to reduce the uncertainty of energy consumption and carbon emissions caused by extreme climate change. In addition, decision-makers may give preferential subsidies to green energy use and reduce the heavy dependence of economic growth on fossil energy consumption with the increase in economic growth in OECD and non-OECD countries. Moreover, the decision-makers may also improve the innovation capacity and energy efficiency and strengthen the R&D, promotion, and application of renewable energy technologies, such as promoting the development and utilization of wind energy and solar energy. In the long term, the income from green technology innovation in the energy field is an effective measure to enhance international competitiveness. Although individual countries may have achieved this goal, it is still lacking on the whole.

One is the limitation of the sample, mainly in OECD and non-OECD countries, and the other is the limitation of the variable. Other variables affecting the climate and environment, such as precipitation, methane, and nitrous oxide, are not taken into account. In future research, this study would be carried out in more countries, such as collecting more information on temperature, precipitation, energy consumption, and carbon emissions of countries in different climate regions and deeply analyzing the relationship between climate change, energy consumption, and carbon emissions. On the basis of improving data availability, other greenhouse gases (such as methane, nitrous oxide, hydrofluorocarbons, perfluorocarbons, and sulfur hexafluoride) besides carbon emissions could more accurately measure environmental pollution. In addition, further analyses of how governments formulate reasonable and effective energy and environmental policies to deal with extreme climate change, improve international competitiveness, and achieve sustainable economic growth.

## Figures and Tables

**Figure 1 ijerph-20-02042-f001:**
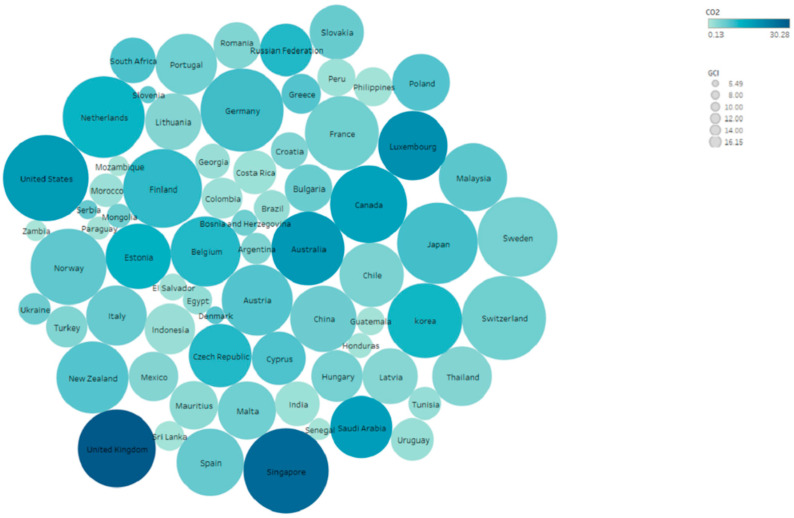
The impact of international competitiveness on CO_2_ emissions varies.

**Figure 2 ijerph-20-02042-f002:**
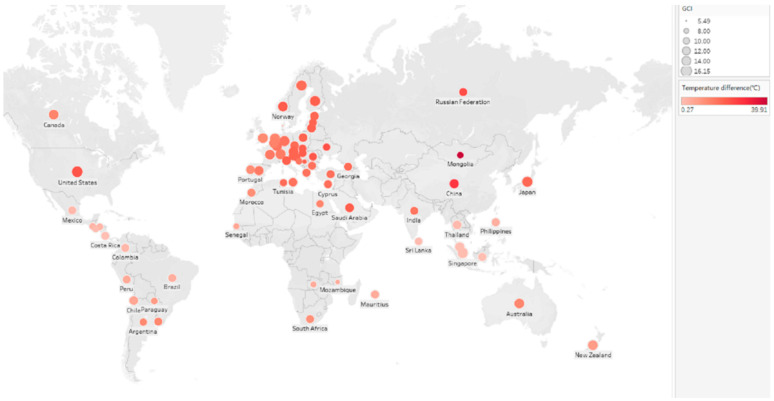
The impact of international competitiveness on temperature difference varies geographically.

**Table 1 ijerph-20-02042-t001:** Variable definition and calculation.

Variables	Abbreviation	Calculation
Dependent variable		
Energy consumption	E	Fossil fuel energy consumption divide by total energy consumption
Climate change	Temp	Absolute value of summer temperature minus winter temperature
Environment pollution	Pol	CO_2_ emissions (metric tons per capita)
Independent variable		
International competitiveness	GCI	Global Competitiveness Index
Economic growth	Seg	Calculation by Principal Components Analysis (PCA)
Control variable		
Climate policy	Poli	The number 1 represents joining the Paris Agreement or Kyoto Protocol and the number 0 represents no joining
Industrialization	Ind	Value added of industry divide by total GDP
Capital	K	Capital stock at constant 2010 national prices
Urbanization	Urb	Urban population divide by the total population

**Table 2 ijerph-20-02042-t002:** Descriptive statistics.

Variables	Group	Obs	Mean	Std. Dev	Min	Max
Pol	OECD	560	1.963	0.64	0.182	4.161
	Non-OECD	576	0.913	1.149	−2.645	3.708
Seg	OECD	560	0.518	0.559	−5.360	1.828
	Non-OECD	576	0.653	0.335	−0.625	1.825
K	OECD	560	2.912	1.361	1.072	14.49
	Non-OECD D	576	1.411	0.972	−2.117	3.633
e	OECD	560	4.499	0.930	0.693	4.585
	Non-OECD D	576	3.898	1.044	0	4.605
GCI	OECD	560	1.910	0.894	1.303	4.450
	Non-OECD D	576	1.736	0.894	0.077	4.440
Ind	OECD	560	3.224	0.253	2.353	3.856
	Non-OECD	576	3.345	0.297	2.301	4.252
Temp	OECD	560	15.53	6.018	0	28.73
	Non-OECD D	576	12.02	9.318	0.01	43.97
Urb	OECD	560	4.367	0.534	−0.202	8.946
	Non-OECD	576	4.045	0.348	2.901	4.605
Poli	OECD	560	0.939	0.242	0	1
	Non-OECD D	576	0.944	0.229	0	1

**Table 3 ijerph-20-02042-t003:** The climate change function (OECD).

Variables	Model (1)	Model (2)	Model (3)	Model (4)
	Sys-GMM	Sys-GMM	Sys-GMM	Sys-GMM
L. Temp	0.0334 **	0.233 ***	0.0144	0.255 ***
	(0.0133)	(0.00633)	(0.0155)	(0.00914)
K	−4.091 ***	0.652 ***	−4.613 ***	0.625 ***
	(0.270)	(0.126)	(0.321)	(0.117)
E	−0.368 ***	0.552 ***	−0.234 **	0.647 ***
	(0.120)	(0.133)	(0.0949)	(0.146)
GCI			0.228 ***	0.106 ***
			(0.0185)	(0.0111)
Constant		8.013 ***		7.154 ***
		(0.404)		(0.553)
Observations	490	525	490	525
Sample	35	35	35	35
AR (1)	0.000	0.000	0.000	0.000
AR (2)	0.305	0.140	0.304	0.175
Hansen test	0.223	0.251	0.163	0.234

Note: Robust standard errors in parentheses, *** *p* < 0.01, ** *p* < 0.05.

**Table 4 ijerph-20-02042-t004:** The climate change function (non-OECD).

Variables	Model (1)	Model (2)	Model (3)	Model (4)
	Sys-GMM	Sys-GMM	Sys-GMM	Sys-GMM
L. Temp	0.147 ***	0.293 ***	0.151 ***	0.551 ***
	(0.0111)	(0.0126)	(0.0110)	(0.0126)
K	0.120	1.416 ***	0.228 *	1.226 ***
	(0.108)	(0.118)	(0.126)	(0.0977)
E	1.247 ***	1.524 ***	1.180 ***	−0.286 ***
	(0.249)	(0.145)	(0.291)	(0.140)
GCI			−0.0633 **	−0.389 ***
			(0.0315)	(0.0190)
Constant		0.253		5.380 ***
		(0.666)		(0.501)
Observations	504	540	504	540
Sample	36	36	36	36
AR (1)	0.000	0.000	0.000	0.000
AR (2)	0.958	0.356	0.907	0.144
Hansen test	0.310	0.356	0.267	0.308

Note: Robust standard errors in parentheses, *** *p* < 0.01, ** *p* < 0.05, * *p* < 0.1.

**Table 5 ijerph-20-02042-t005:** The energy consumption function (OECD).

Variables	Model (1)	Model (2)	Model (3)	Model (4)	Model (5)	Model (6)	Model (7)	Model (8)
	Sys-GMM	Sys-GMM	Sys-GMM	Sys-GMM	Sys-GMM	Sys-GMM	Sys-GMM	Sys-GMM
L. E	0.785 ***	0.962 ***	1.013 ***	0.981 ***	1.115 ***	0.870 ***	1.512 ***	0.858 ***
	(0.102)	(0.0454)	(0.113)	(0.0424)	(0.125)	(0.0644)	(0.193)	(0.0709)
Temp	0.00943 **	−0.00779 ***	0.00168	−0.00888 **	−0.00286	0.00176	−0.0136 *	0.00138
	(0.00461)	(0.00293)	(0.00455)	(0.00423)	(0.00468)	(0.00309)	(0.00765)	(0.00356)
Ind	0.635 ***	0.0509	0.390 ***	0.0793	−0.277	−0.0336	−0.660 *	−0.0293
	(0.141)	(0.0375)	(0.140)	(0.0485)	(0.329)	(0.0709)	(0.377)	(0.0574)
GCI			0.00899	0.00799			0.0200	0.000378
			(0.0113)	(0.00916)			(0.0142)	(0.00879)
Seg					−0.256 ***	−0.213 ***	−0.365 **	−0.187 ***
					(0.0962)	(0.0559)	(0.144)	(0.0575)
Constant		0.111		−0.0530		0.674 *		0.706 **
		(0.231)		(0.212)		(0.354)		(0.321)
Observations	490	525	490	525	490	525	490	525
Sample	35	35	35	35	35	35	35	35
AR (1)	0.0116	0.0107	0.00995	0.0106	0.0126	0.00792	0.0141	0.00920
AR (2)	0.255	0.374	0.334	0.372	0.385	0.305	0.496	0.303
Hansen test	0.408	0.101	0.285	0.108	0.660	0.373	0.726	0.290

Note: Robust standard errors in parentheses, *** *p* < 0.01, ** *p* < 0.05, * *p* < 0.1.

**Table 6 ijerph-20-02042-t006:** The energy consumption function (non-OECD).

Variables	Model (1)	Model (2)	Model (3)	Model (4)	Model (5)	Model (6)	Model (7)	Model (8)
	Sys-GMM	Sys-GMM	Sys-GMM	Sys-GMM	Sys-GMM	Sys-GMM	Sys-GMM	Sys-GMM
L. E	−0.158	0.603 ***	−0.234 **	0.581 ***	−0.257 **	0.603 ***	−0.382 ***	0.587 ***
	(0.116)	(0.0299)	(0.107)	(0.0322)	(0.116)	(0.0293)	(0.0959)	(0.0313)
Temp	0.00119	−0.00200	−0.000769	−0.00360	0.00516	−0.000842	0.00489	−0.00308
	(0.00579)	(0.00333)	(0.00583)	(0.00363)	(0.00608)	(0.00296)	(0.00655)	(0.00333)
Ind	0.790 ***	0.0999 *	0.947 ***	0.112 **	0.680 ***	0.101	0.847 ***	0.161 *
	(0.201)	(0.0544)	(0.238)	(0.0571)	(0.225)	(0.0747)	(0.259)	(0.0912)
GCI			−0.00898 **	−0.00649 **			−0.0100 **	−0.00712 **
			(0.00439)	(0.00298)			(0.00436)	(0.00328)
Seg					−0.0994	−0.0204	−0.110 *	0.0159
					(0.0676)	(0.0387)	(0.0597)	(0.0487)
Constant		1.297 ***		1.375 ***		1.302 ***		1.177 ***
		(0.167)		(0.179)		(0.252)		(0.313)
Observations	504	540	504	540	504	540	504	540
Sample	36	36	36	36	36	36	36	36
AR (1)	0.842	0.0525	0.854	0.0550	0.760	0.0498	0.347	0.0505
AR (2)	0.208	0.568	0.192	0.571	0.190	0.576	0.189	0.565
Hansen test	0.0764	0.160	0.0565	0.120	0.0763	0.141	0.0441	0.125

Note: Robust standard errors in parentheses, *** *p* < 0.01, ** *p* < 0.05, * *p* < 0.1.

**Table 7 ijerph-20-02042-t007:** The environmental pollution function (OECD).

Variables	Model (1)	Model (2)	Model (3)	Model (4)	Model (5)	Model (6)	Model (7)	Model (8)	Model (9)	Model (10)
	Sys-GMM	Sys-GMM	Sys-GMM	Sys-GMM	Sys-GMM	Sys-GMM	Sys-GMM	Sys-GMM	Sys-GMM	Sys-GMM
L. Pol	0.686 ***	0.862 ***	0.636 ***	0.937 ***	0.705 ***	0.938 ***	0.672 ***	0.939 ***	0.651 ***	0.940 ***
	(0.00739)	(0.00891)	(0.00945)	(0.00966)	(0.0156)	(0.0122)	(0.0137)	(0.0117)	(0.0201)	(0.0141)
Temp	−0.00689 ***	−0.00597	0.000872	0.00653 *	5.88 × 10^−5^	−0.00233	0.00998	−0.00331	0.00186	−0.00625
	(0.00262)	(0.00746)	(0.00333)	(0.00380)	(0.00596)	(0.00389)	(0.00657)	(0.00452)	(0.00675)	(0.00501)
Temp2	0.000720 ***	0.000590 ***	0.000530 ***	0.000286 ***	0.000547 ***	0.000525 ***	0.000312 *	0.000555 ***	0.000483 ***	0.000602 ***
	(0.000073)	(0.000203)	(0.000099)	(0.000098)	(0.000159)	(0.000103)	(0.000166)	(0.000117)	(0.000162)	(0.000144)
E	−0.0117 ***	0.0261 ***	−0.0270 ***	0.00373	0.0102 ***	0.0101 ***	0.0203 ***	0.00655 *	−0.0168 **	0.0112 ***
	(0.00329)	(0.00211)	(0.00588)	(0.00277)	(0.00236)	(0.00315)	(0.00556)	(0.00336)	(0.00670)	(0.00416)
GCI			−0.0147 ***	−0.00777 ***			−0.0155 ***	−0.00626 ***	−0.0158 ***	−0.00628 ***
			(0.00109)	(0.000897)			(0.000759)	(0.000551)	(0.000678)	(0.000645)
Seg					0.0111 *	0.000575	0.0208 ***	0.00228	0.0148 **	0.00487
					(0.00632)	(0.00494)	(0.00675)	(0.00575)	(0.00585)	(0.00555)
Urb									0.00634	0.0124
									(0.0162)	(0.00807)
Poli										0.0131
										(0.0115)
Constant		0.0992		−0.0616 **		−0.0277		−0.00139		−0.0543
		(0.0631)		(0.0302)		(0.0392)		(0.0417)		(0.0712)
Observations	490	525	490	525	490	525	490	525	490	525
Sample	35	35	35	35	35	35	35	35	35	35
AR (1)	0.191	0.192	0.188	0.192	0.193	0.193	0.191	0.193	0.196	0.194
AR (2)	0.748	0.609	0.838	0.616	0.739	0.607	0.811	0.611	0.794	0.615
Hansen test	0.735	0.841	0.708	0.843	0.714	0.884	0.682	0.846	0.687	0.864

Note: Robust standard errors in parentheses, *** *p* < 0.01, ** *p* < 0.05, * *p* < 0.1.

**Table 8 ijerph-20-02042-t008:** The environmental pollution function (non-OECD).

Variables	Model (1)	Model (2)	Model (3)	Model (4)	Model (5)	Model (6)	Model (7)	Model (8)	Model (9)	Model (10)
	Sys-GMM	Sys-GMM	Sys-GMM	Sys-GMM	Sys-GMM	Sys-GMM	Sys-GMM	Sys-GMM	Sys-GMM	Sys-GMM
L. Pol	0.903 ***	0.911 ***	0.886 ***	0.953 ***	0.839 ***	0.944 ***	0.831 ***	0.939 ***	0.548 ***	0.887 ***
	(0.00735)	(0.00677)	(0.00571)	(0.00475)	(0.00969)	(0.00717)	(0.00666)	(0.00812)	(0.0180)	(0.0105)
Temp	−0.0526 ***	−0.0222 ***	−0.0524 ***	−0.0277 ***	−0.0441 ***	−0.0247 ***	−0.0422 ***	−0.0236 ***	−0.0430 ***	−0.0293 ***
	(0.00179)	(0.00347)	(0.00320)	(0.00393)	(0.00262)	(0.00472)	(0.00452)	(0.00398)	(0.00467)	(0.00435)
Temp2	0.00130 ***	0.000623 ***	0.00130 ***	0.000766 ***	0.00111 ***	0.000681 ***	0.00107 ***	0.000664 ***	0.00113 ***	0.000802 ***
	(5.39 × 10^−5^)	(7.92 × 10^−5^)	(8.56 × 10^−5^)	(9.07 × 10^−5^)	(6.30 × 10^−5^)	(0.000102)	(0.000111)	(9.53 × 10^−5^)	(0.000118)	(0.000102)
E	0.0248 ***	0.0601 ***	0.0259 ***	0.0658 ***	0.0394 ***	0.0719 ***	0.0404 ***	0.0824 ***	0.0964 ***	0.0969 ***
	(0.00402)	(0.00350)	(0.00479)	(0.00809)	(0.00974)	(0.00851)	(0.00714)	(0.0133)	(0.0166)	(0.0151)
GCI			0.00418 ***	0.00327 ***			0.00370 ***	0.00232 **	−0.00780 ***	0.00303 ***
			(0.000649)	(0.000807)			(0.000476)	(0.000940)	(0.00124)	(0.00112)
Seg					0.0692 ***	0.0384 ***	0.0685 ***	0.0460 ***	0.0292 **	0.0347 ***
					(0.00575)	(0.00401)	(0.00556)	(0.00534)	(0.0130)	(0.00747)
Urb									1.782 ***	0.210 ***
									(0.172)	(0.0682)
Poli										−0.0705 ***
										(0.00933)
Constant		−0.0125		−0.0521		−0.0998 **		−0.164 **		−0.900 ***
		(0.0312)		(0.0440)		(0.0498)		(0.0637)		(0.281)
Observations	504	540	504	540	504	540	504	540	504	540
Sample	36	36	36	36	36	36	36	36	36	36
AR (1)	0.0215	0.0390	0.0219	0.0367	0.0211	0.0362	0.0217	0.0372	0.0279	0.0332
AR (2)	0.436	0.719	0.447	0.709	0.661	0.833	0.679	0.911	0.998	0.935
Hansen test	0.712	0.840	0.712	0.828	0.708	0.850	0.659	0.880	0.632	0.860

Note: Robust standard errors in parentheses, *** *p* < 0.01, ** *p* < 0.05.

**Table 9 ijerph-20-02042-t009:** Time heterogeneity analysis (OECD).

Time	Variables	Climate Change		Energy Consumption		Environmental Pollution	
		Diff-GMM	Sys-GMM	Diff-GMM	Sys-GMM	Diff-GMM	Sys-GMM
2015–2019	GCI	0.675 *	0.104	−0.0094 *	−0.0027 *	0.000600	−0.00819 ***
		(0.362)	(0.0772)	(0.0317)	(0.0106)	(0.00315)	(0.00316)
	Seg			0.0419	0.0156	0.0132	−0.00842
				(0.016)	(0.059)	(0.110)	(0.0289)
2004–2019	GCI	0.228 ***	0.106 ***	0.0200	0.000378	−0.0158 ***	−0.00628 ***
		(0.0185)	(0.0111)	(0.0142)	(0.00879)	(0.000678)	(0.000645)
	Seg			−0.365 **	−0.187 ***	0.0148 **	0.00487
				(0.144)	(0.0575)	(0.00585)	(0.00555)

Note: Robust standard errors in parentheses, *** *p* < 0.01, ** *p* < 0.05, * *p* < 0.1.

**Table 10 ijerph-20-02042-t010:** Time heterogeneity analysis (non-OECD).

Time	Variables	Climate Change		Energy Consumption		Environmental Pollution	
		Diff-GMM	Sys-GMM	Diff-GMM	Sys-GMM	Diff-GMM	Sys-GMM
2015–2019	GCI	−0.595	−0.128 *	−0.0078 *	−0.0064 *	−0.00832	0.00699 **
		(0.415)	(0.0723)	(0.0293)	(0.0316)	(0.00530)	(0.00284)
	Seg			0.0214	0.0156	−0.212	0.0692 **
				(0.0314)	(0.059)	(0.259)	(0.0346)
2004–2019	GCI	−0.0633 **	−0.389 ***	−0.0100 **	−0.00712 **	−0.0078 ***	0.00303 ***
		(0.0315)	(0.0190)	(0.00436)	(0.00328)	(0.00124)	(0.00112)
	Seg			−0.110 *	0.0159	0.0292 **	0.0347 ***
				(0.0597)	(0.0487)	(0.0130)	(0.00747)

Note: Robust standard errors in parentheses, *** *p* < 0.01, ** *p* < 0.05, * *p* < 0.1.

## Data Availability

The data presented in this study are available on request from the corresponding author. The data are not publicly available due to privacy or ethical restrictions.

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
