# Peer review of "The Dynamic Nexus of Fossil Energy Consumption, Temperature and Carbon Emissions: Evidence from Simultaneous Equation Model"

_ijerph, 2023, doi:10.3390/ijerph20032042_

Round 1

Reviewer 1 Report

The manuscript discusses the dynamics of energy consumption, temperature and carbon emissions. And it compares OECD countries with non-OECD countries. The manuscript theme has some meaning but is poorly described. Lack of essential comparison with other studies. Major changes are recommended.

Abstract

1. Discourse repetition before and after part of the abstract. Rearrange is recommended. I suggest combing the background of the article, the methods used, the conclusions, and the significance of the facts. Moreover, the conclusions of the literature review in the paper are repeatedly emphasized in the abstract whether it is reasonable or not.

Introduction and Literature review

2. The current introduction and literature review are unreasonable. The introductory paragraph is too long. The author should divide it up more properly. The introduction introduces the background, the research area, and the necessity of the research. Literature reviews need to reflect similar regional studies and methodological comparisons. Finally, the purpose of the research is presented.

Methods

3. Is it appropriate to place data sources in Section 4.1? It is recommended that data sources be placed in the method presentation section.

Results

4. The result discussion section of the article is very subjective. The data are scarce. Rearrange is recommended. It is suggested that the results show the particularity of the data. Separate the discussion from the results. Increase the comparison with other studies. To prove the scientific nature and particularity of the article.

5. The article mentions that the temperature difference and emission are U-shaped. It is more convenient for readers to understand intuitively with the form of pictures. And compare the two areas.

6. Revise the grammar of the manuscript. Avoid such expressions as this paper. Read more about the study.

7. The manuscript aims to explore the relationship between energy, temperature, and carbon emissions. But the article describes the relationship between the three as relatively independent. The author does not have a comprehensive discussion and suggests rectification.

Conclusion

8. The conclusion is broad. It is suggested to make a comparative analysis with the results and discussion to summarize the important conclusions. Rather than displaying commonsense advice.

Reviewer 3 Report

Overall, the manuscript was able to develop the study in the context of global environmental protection, was well intentioned, well read, and technically sound. I would recommend acceptance of the manuscript for publication in ijerph.

but some issues should be corrected before considering publication. Please correct the following issues:

1.The introduction needs to be appropriately streamlined. Please add the latest literature, such as: https://doi.org/10.3390/ijerph192013385;

2.The conclusion section lacks a summary of the findings. It is suggested that the conclusions of the paper should be presented clearly.

3.The figure is not clear enough. And the manuscript uses carbon dioxide to represent greenhouse gases, and this needs to be considered.

4. It is suggested to put forward specific policy recommendations on the basis of the conclusions of this manuscript.

Round 2

Reviewer 1 Report

The author has systematically modified chapter structure and viewpoint expression. There has been a marked improvement in the quality of the manuscript, but further revisions are needed to meet publication standards.

1. Is the position of Figure 1 reasonable? Figure 1 shows the relationship between international competitiveness and temperature influence. But it is in Chapter 2 that you first mention how to quantify international competitiveness.

2. The indicators involved are mentioned in the manuscript on international competitiveness.  But there is no clear method to introduce and reference. As a factor of innovation, it is necessary to explore its quantitative standard.

3.  Pay attention to the English expression of the manuscript. It is recommended to check the whole text, such as, line 195.

4. Note references in the manuscript, such as line 308 data sources.

5. The author should discuss and research more systematically, how is your work similar and different from the work of others?
